# Th17-associated cytokines IL-17 and IL-23 in inflamed skin of Darier disease patients as potential therapeutic targets

Monika Ettinger[1,2], Teresa Burner[2], Anshu Sharma [3], Yun-Tsan Chang[4], Angelika Lackner[2], Pacôme Prompsy [4], Isabella M. Deli [1], Judith Traxler[1], Gerald Wahl[1], Sabine Altrichter [1,2], Rupert Langer[5,6], Yi-Chien Tsai[4], Suraj R. Varkhande [3], Leonie C. Schoeftner[3], Christoph Iselin[4], Iris K. Gratz[3], Susanne Kimeswenger [2,8], Emmanuella Guenova [4,7,8] & Wolfram Hoetzenecker [1,2,8] ✉

Darier disease (DD) is a rare, inherited multi-organ disorder associated with mutations in the *ATP2A2* gene. DD patients often have skin involvement characterized by malodorous, inflamed skin and recurrent, severe infections. Therapeutic options are limited and inadequate for the long-term management of this chronic disease. The aim of this study was to characterize the cutaneous immune infiltrate in DD skin lesions in detail and to identify new therapeutic targets. Using gene and protein expression profiling assays including scRNA sequencing, we demonstrate enhanced expression of Th17-related genes and cytokines and increased numbers of Th17 cells in six DD patients. We provide evidence that targeting the IL-17/IL-23 axis in a case series of three DD patients with monoclonal antibodies is efficacious with significant clinical improvement. As DD is a chronic, relapsing disease, our findings might pave the way toward additional options for the long-term management of skin inflammation in patients with DD.

Darier Disease (DD) or dyskeratosis follicularis is a rare autosomal dominant inherited disorder (prevalence: 1–3:100,000) defined by heterozygous mutations in the *ATP2A2* gene encoding the sarcoendoplasmic reticulum Ca$^{2+}$ ATPase isoform 2 (SERCA2)[1]. In addition to an increased risk of neuropsychiatric disorders, type 1 diabetes, and heart failure, the skin is the most commonly affected organ in DD patients[2–8]. DD manifests as cutaneous keratotic papules and malodorous plaques, often affecting large areas of the body and significantly affecting patients' quality of life[9]. The disease has a late onset, often around puberty, and a chronic course with exacerbations triggered by sun exposure, heat, friction, or infection, often requiring hospitalization with intravenous treatment (e.g., antibiotic and/or antiviral drugs)[3]. Therapeutic options are limited and inadequate for the long-term management of this chronic disease with recurrent severe bacterial and viral infections of the skin[9,10]. Conventional treatment still relies on the short-term use of topical corticosteroids, antiseptics, and systemic antibiotics. Currently, the most effective treatment is systemic retinoids, but their use is limited by side effects[10,11]. The effects of the disease-causing mutations in the *ATP2A2* gene are well studied in the epidermis, where they result in aberrant

[1]Department of Dermatology and Venereology, Kepler University Hospital Linz, Linz, Austria. [2]Department of Dermatology and Venereology, Medical Faculty, Johannes Kepler University Linz, Linz, Austria. [3]Department of Biosciences and Molecular Biology, University of Salzburg, Salzburg, Austria. [4]Department of Dermatology, University of Lausanne and Faculty of Biology and Medicine, Lausanne, Switzerland. [5]Institute of Pathology and Molecular Pathology, Kepler University Hospital Linz, Linz, Austria. [6]Institute of Pathology and Molecular Pathology, Medical Faculty, Johannes Kepler University Linz, Linz, Austria. [7]Department of Dermatology, Hospital 12 de octubre, Medical school, University Complutense, Madrid, Spain. [8]These authors contributed equally: Susanne Kimeswenger, Emmanuella Guenova, Wolfram Hoetzenecker. ✉e-mail: wolfram.hoetzenecker@kepleruniklinikum.at

Ca$^{2+}$ signaling, loss of intercellular connections (acantholysis), and apoptosis of keratinocytes[12–15]. By contrast, the consequences of *ATP2A2* gene mutations in other cell types, particularly immune cells, are poorly understood[16]. Because chronic inflammation of the skin is common and deleterious in DD, we hypothesized that the involvement of the skin's immune system is important in the pathogenesis of DD and aimed to characterize the cellular and molecular composition of the cutaneous immune infiltrate in DD skin lesions.

Here we show that IL-17 signaling and T helper type 17 (Th17) cells are increased in the lesional skin of DD patients. Treatment with anti-IL-17 or anti-IL-23 antibodies improves clinical symptoms and is associated with a normalization of the cytokine profiles in a case series of three DD patients.

## Results

We included six patients with DD in our cohort. DD was verified by histology of skin biopsies and genetic analyses. The characteristics and previous therapies of the patients (3 males, 3 females; age range 20–60 years) are summarized in Table 1. The patients had skin manifestations since adulthood, ranging from moderate (DD patient number 4 [PAT4]) to severe (PAT1, 2, 5, 7, 9) forms of disease with recurrent bacterial and viral cutaneous infections, some of which required hospitalization (Table 1, Suppl. Fig. 1). All patients had been previously treated with various drugs that provided no or only short-term clinical benefit (Table 1). Using single cell RNA sequencing (scRNA-seq), we profiled cells from skin biopsies of DD lesions, which we compared with publicly available scRNA-seq data from psoriasis and normal healthy skin (Fig. 1a, Suppl. Fig. 2)[17]. Based on cluster analysis, 13 cell populations with distinct expression profiles were defined (Suppl. Fig. 2). Cell populations were found in all three groups, with increased numbers of T cells in both DD and psoriasis samples consistent with the inflammatory nature of these diseases (Fig. 1a, Suppl. Fig. 2). Gene expression analysis and subsequent gene set enrichment analysis (GSEA) of DD keratinocytes revealed overexpression of genes involved in mTORC1 signaling and p53 pathway when compared with keratinocytes from normal, healthy skin (Suppl. Fig. 3, Suppl. Table 1). Because an important clinical aspect of DD is chronic skin inflammation, we next performed high-throughput immunoprofiling using NanoString technology (Fig. 1b) of immune-related genes expressed in the lesional skin of six DD patients. GSEA of the gene expression data revealed that IL-17-signaling was enhanced in the skin of DD patients compared with the skin of healthy controls (HC; Fig. 1b, c, Suppl. Table 2). Interestingly, the expression profile resembled that of patients with psoriasis, which is known to be a Th17 driven disease (Fig. 1b, Suppl. Table 3)[18]. Based on these results, we examined Th17-, Th1-, and Th2-related cytokines in DD patients by qRT-PCR and found significantly increased *IL17A* expression compared to HC skin (Fig. 1d, Suppl. Fig. 4). Specifically, five of six DD patients showed increased expression of *IL17A* and one patient displayed increased expression of *IL23A* (Suppl. Fig. 4, Table 1). Enhanced expression of IL-17A and IL-22, which is produced by several populations of immune cells at a site of inflammation, including Th17 cells, was confirmed at the protein level by bead-based immunoassays of the protein extract of DD skin biopsies compared with HC skin (Fig. 1e).

IL-17(A-F) are pro-inflammatory cytokines produced mainly by a group of T helper (Th) cells known as Th17 cells. Th17 cells play a major role in the pathogenesis of various autoimmune diseases (e.g., rheumatoid arthritis, inflammatory bowel disease, and psoriasis)[19]. To identify the origin of IL-17 in the lesional skin of DD patients, we analyzed *IL17* gene expression at the single cell level in our scRNA-seq data set. Among the different cell compartments/types, we detected *IL17A/F* expression predominantly in the Th cell subset in skin samples of DD and psoriasis (Fig. 2a) suggesting that Th17 cells are the main source of IL-17 in inflamed skin of patients with DD. The presence of the Th17-

**Table 1 | Darier Disease patient characteristics and treatments**

| Patient No./ age/sex | Heterogenous ATP2A2 mutation | Phenotype/disease severity[a] | Complications | Prior therapies | Increased cytokine[b] | Current therapy |
|---|---|---|---|---|---|---|
| PAT1/26-30/f | c.215 C > A, (p.Ser72Tyr) | Severe | recurrent bacterial cutaneous superinfections | topical steroids, antiseptics, systemic therapy with iso-tretinoin 06/18-02/19 (no beneficial effect) | IL23A | systemic therapy with guselk-umab since 11/21 |
| PAT2/41-45/m | c.1184 T > G, (p.Val395Gly) | Severe | recurrent bacterial cutaneous superinfections | topical steroids, antiseptics, systemic therapy with aci-tretin 08/17-09/17, 04/21–09/21 (no benefit), CO$_2$ laser on legs (temporary benefit) | IL17A | systemic therapy with secuki-numab since 12/21 |
| PAT4/46-50/m | c.2305 G > A (p.Gly769Arg) | Moderate | recurrent HSV infections and bacterial cutaneous infections | topical steroids, antiseptics, systemic therapy with aci-tretin 12/18-07/19 (no benefit), CO$_2$ laser on legs (temporary benefit) | IL17A | topical steroids, antiseptics |
| PAT5/21-25/f | c.392 G > A p.(Arg131Gln) | Severe | recurrent bacterial cutaneous superinfections | no topical or systemic therapy | IL17A | loss of follow up |
| PAT7/56-60/m | c.224delT p.(Leu75Trpfs*15) | Severe | recurrent bacterial cutaneous superinfections | topical steroids, antiseptics | IL17A | systemic therapy with aci-tretin since 09/22 |
| PAT9/46-50/f | c.1287+1delG splice mutation | Severe | recurrent bacterial cutaneous superinfections depression | topical steroids, antiseptics, systemic therapy with isotretinoin 01/08-09/22 (temporary benefit) | IL17A | systemic therapy with secuki-numab since 11/22 |

[a]No standardized assessment scale has been validated to date for DD. The disease was considered severe if the patient had a body surface area (BSA) > 35%, moderate if BSA was 10–35%.
[b]Cytokine that has been shown to be increased in the patient's lesional skin.

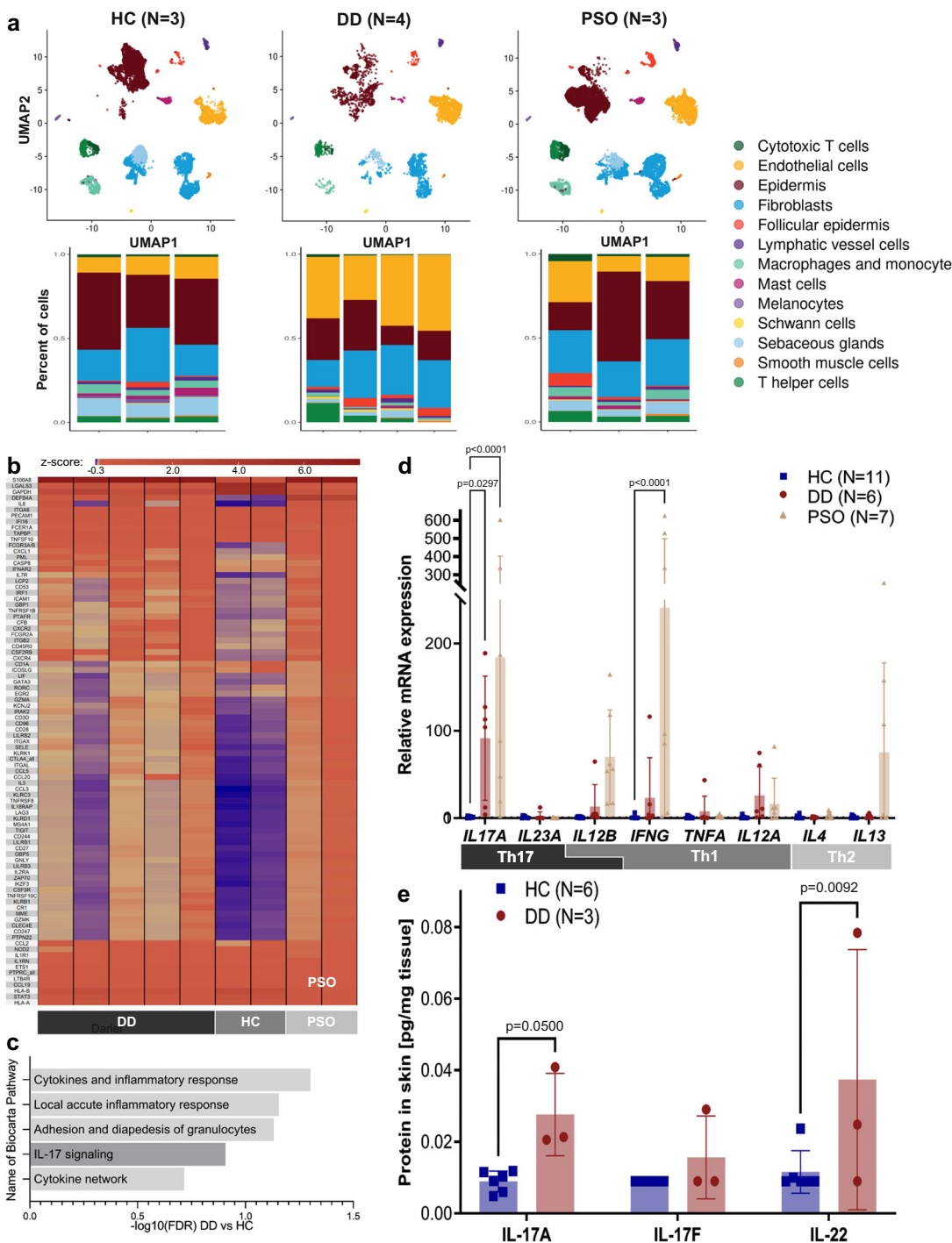

**Fig. 1 | Enhanced expression of Th17-related genes and cytokines in the skin of patients with Darier disease (DD). a** UMAP analysis (upper panel) and percentage of different cell types (lower panel) determined by scRNA-seq analysis of skin samples from four DD patients (lesional skin, N = 4, 3746 cells). Results were compared to publicly available scRNA-seq data of psoriasis (PSO, lesional skin, N = 3, 11417 cells) and healthy controls (HC, N = 3, 12817 cells)[17]. **b** Heat map of differentially expressed (DE) genes expressed as Z-score (DD vs HC, fold change > 1.5) as determined by NanoString nCounter analysis using the Immunology_V2 panel in skin of untreated DD patients (N = 5), healthy controls (N = 2) and psoriasis patients (N = 2). **c** Biocarta pathways most enriched in DD patients as compared to HC, as determined by NanoString nCounter and Gene Set Enrichment Analysis. Pathways are ordered according to their normalized enrichment scores. Bars represent -log10 of False Discovery Rates (FDR) (See Suppl. Table 2 for complete table). **d** Relative mRNA expression (qRT-PCR) of Th1, Th2, and Th17-associated cytokines normalized to housekeeping gene ACTB and relative to HC. Bars represent means, error bars represent standard deviations. p-values were calculated using 2-way ANOVA with Dunnett multiple comparison correction (all against HC). **e** Protein expression of Th17-related cytokines in the skin of HC and DD patients as determined by bead-based immunoassays (LegendPlex analysis). p-values were calculated using 2-way ANOVA. Error bars represent standard deviation.

promoting cytokine IL-23 in keratinocytes (Fig. 2b, c) as well as the presence of Th17 cells (Fig. 2d, e) in DD was confirmed by multi-color immunofluorescence (IF) stainings of tissue microarray (TMA) sections (Fig. 2b–e, Suppl. Fig. 5) and conventional IF stainings of paraffin sections (Suppl. Fig. 6). Quantification of TMA stainings revealed increased numbers of Th17 cells (CD4⁺IL-17⁺) in DD skin samples compared with skin lesions of patients with atopic dermatitis (AD), which is known to be a classical Th2-(IL-4/IL-13) dominated skin disease

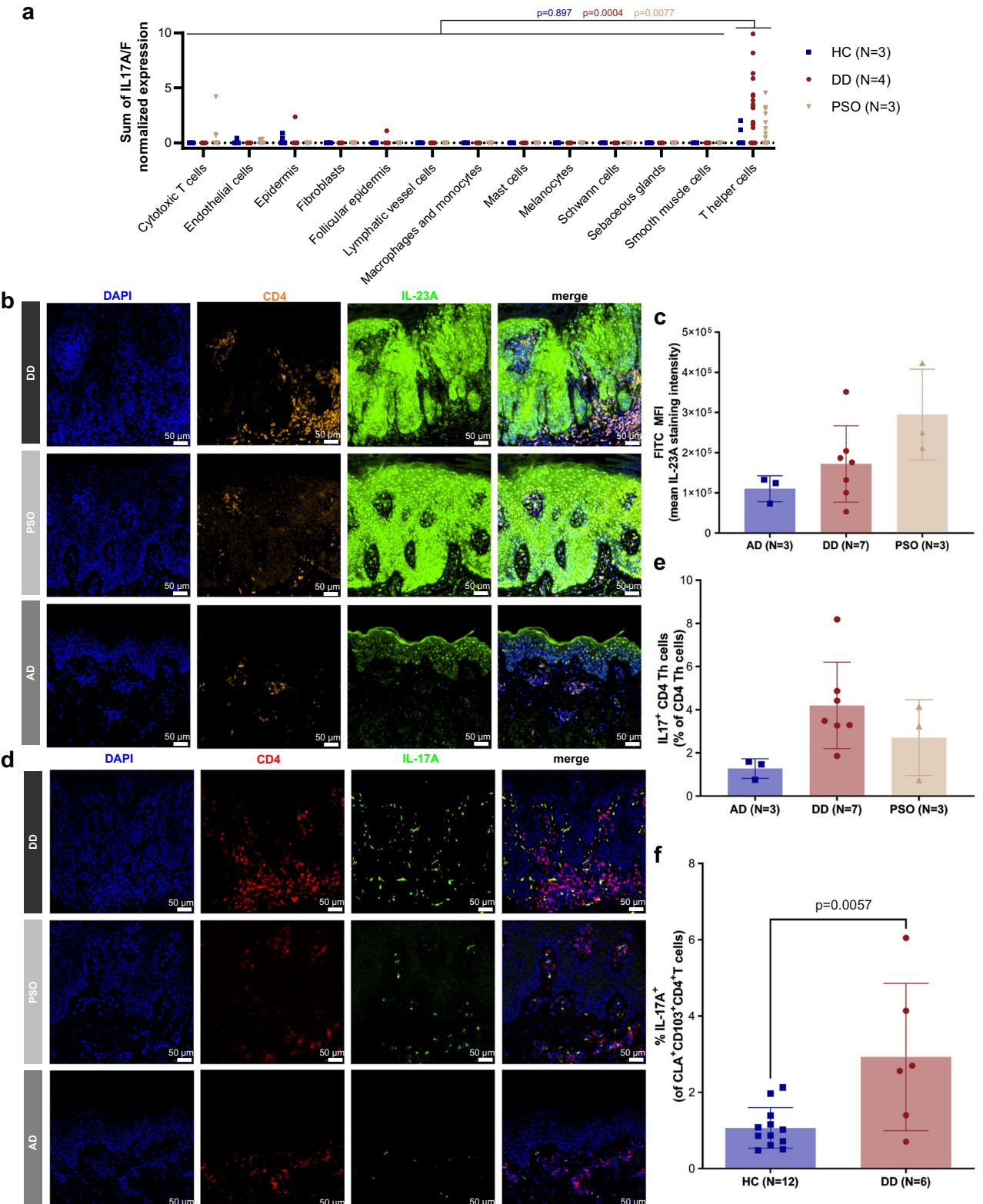

(Fig. 2e)[20]. Likewise, quantification of mean fluorescence intensity of IL-23A staining revealed increased expression of IL-23 in the skin of DD and PSO patients as compared to AD patients (Fig. 2c). In line with these tissue-data, flow-cytometric analysis of PBMC revealed increased numbers of IL-17A-producing CD103+CLA+ circulating skin-resident memory CD4+ T cells (cT$_{RM}$) in the blood of patients compared to HC (Fig. 2f, Suppl. Fig. 7). cT$_{RM}$ are cutaneous T$_{RM}$ cells that have exited the skin and can be found in the circulation. In consequence, these cells share cytokine profile with skin T$_{RM}$ and can therefore be utilized to analyze skin T$_{RM}$ from the blood[21].

Because our results indicated an enhanced IL-17/IL-23 axis in inflamed DD skin, we took a precision medicine approach and administered monoclonal antibodies targeting the overexpressed cytokine to treatment-refractory patients. Specifically, we treated PAT1 with an IL-23A blocking antibody (guselkumab) and PAT2 and PAT9 with an IL-17A blocking antibody (secukinumab). In all three patients, we

**Fig. 2 | Increased numbers of Th17 cells in lesional skin of DD patients. a** Sum of IL17A/F normalized expression in different cell types of healthy controls (HC, $N = 3$, 12817 cells), Darier Disease patients (DD, $N = 4$, 3746 cells) and Psoriasis patients (PSO, $N = 3$, 11417 cells) skin samples as determined by scRNA-seq analysis. Significance bars represent results of one-tailed student's T-test of IL17A/F expression between T helper and non-T helper cells within one group (blue *p*-value for HC, red *p*-value for DD, beige *p*-value for PSO), without adjustment for multiple comparisons. **b** Representative multi-color immunofluorescence (IF) images of tissue micro array sections stained with OPAL technique for IL-23A, CD4 and DAPI in skin samples of DD patients ($N = 7$), atopic dermatitis patients (AD, $N = 3$) and PSO ($N = 3$). CD4 positive T cells are marked in red, IL-23A positive cells are marked in green, DAPI-stained nuclei are marked in blue. Isotype controls are shown in Suppl. Fig. 5. **c** Quantification of IL-23A positive cells as determined by mean fluorescence intensity of multi-color immunofluorescence stained TMA sections (skin samples:

DD ($N = 7$, red), PSO ($N = 3$, beige), AD ($N = 3$, blue)). **d** Representative multi-color immunofluorescence (IF) images of tissue micro array sections stained with OPAL technique for IL-17A, CD4 and DAPI in skin samples of DD patients (N = 7), atopic dermatitis patients (AD, $N = 3$) and PSO ($N = 3$). CD4 positive T cells are marked in red, IL-17A positive cells are marked in green, DAPI-stained nuclei are marked in blue. Isotype controls are shown in Suppl. Fig. 5. **e** Quantification of IL17A/CD4/DAPI triple-positive cells of multi-color immunofluorescence stained TMA sections (skin samples: DD ($N = 7$, red), PSO ($N = 3$, beige), AD ($N = 3$, blue)). **f** IL-17A production of live gated human CD3⁺CD4⁺CD45RA⁻CLA⁺CD103⁺ circulating skin-resident memory T cells ($cT_{RM}$) in the blood of DD patients (red) compared to HC (blue). Flow cytometric analysis of IL-17A upon ex vivo stimulation with PMA/ionomycin and intracellular cytokine staining. *p*-values were calculated using unpaired, two-tailed student's t test. Gating strategy is shown in Suppl. Fig. 7.

observed a rapid decrease in skin inflammation, followed by a reduction and flattening of hyperkeratotic papules and plaques, especially in the most severely affected areas of the body (i.e., the thorax; Fig. 3a, Suppl. Fig. 8). The assessment of clinical scores showed a 50% reduction of the Investigator's Global Assessment (IGA) from 4 points (severe disease) at baseline to 2 points (mild disease) during treatment (Fig. 3b). In addition, we observed a reduction in itching and an improvement in the patients' quality of life (Suppl. Fig. 9). The treatment is well tolerated and has been used in two patients (PAT1, 2) for more than a year. Interestingly, three months after treatment initiation in PAT1 and PAT2, improvement in clinical symptoms was accompanied by normalization of the inflammatory gene expression profile (NanoString and qRT-PCR; Fig. 4a–c) in the patients' skin. In detail, antibody therapy specifically suppressed Th17 associated cytokines (*IL23A* in PAT1, *IL17A* in PAT2) in skin samples, which correlated with a normalization and/or increased expression of Th1 and Th2 cytokines compared to HC (Fig. 4a, b). The decrease in *IL23* expression (Fig. 4a) correlated well with a decrease in Th17 cells in PAT1 skin samples collected before and during antibody therapy (Fig. 4e, Suppl. Fig. 10).

## Discussion

In our patient cohort, we demonstrated increased expression of Th17-related genes and increased numbers of Th17 cells in the skin of DD patients. Based on our observation, we targeted the IL-17/IL-23 axis in a case series of three patients, resulting in an effective and safe therapy.

Treatment of genodermatoses in general is challenging[22]. Because of the small patient population, it is almost impossible to conduct clinical trials and thus get drugs approved. There is a very limited number of effective treatments for DD, and there are no randomized, placebo-controlled trials[9]. Studies addressing the pathophysiology of DD have mainly focused on the epidermis, specifically keratinocytes. In this context, recent reports have shown that mutations in the *ATP2A2* gene lead to impaired calcium homeostasis in keratinocytes, decreased cell-cell adhesion in the epidermis and apoptosis in keratinocytes[3]. These mechanisms are thought to lead to the characteristic skin phenotype of DD with hyperkeratotic papules and plaques and provide further rationale for the use of keratolytic retinoids in the long-term management of patients. Indeed, systemic retinoids (e.g., isotretinoin) are effective in DD patients but are often discontinued due to side effects and the persistence of skin inflammation with recurrent infections[11]. In a recent study, Zaver et al. observed increased MAPK signaling in an organotypic in vitro model of DD with human keratinocytes lacking SERCA2 and identified MEK inhibition as a potential treatment strategy for DD[23].

A hallmark of DD is chronic inflammation of the skin, which is common and deleterious in most patients. Interestingly, reports on the role of the immune system in the pathogenesis and progression of DD are rare. To date, skin inflammation in DD patients has mainly been considered an indirect, secondary effect of impaired epidermal barrier function and/or dysbiosis of the cutaneous microbiome in DD patients.

In a recent report by Chen et al., mutations in the *ATP2A2* gene were for the first time directly associated with impaired immune system function, particularly the B cell compartment[24]. The authors showed that SERCA2 is required for V(D)J recombination and subsequent B cell maturation. Interestingly, some DD patients have reduced numbers of mature B cells in the blood[24]. In our study, we observed an increased number of Th17 cells in the circulation and skin of DD patients, which is comparable to the cell numbers found in psoriasis. Our observations are in line with two recent publications by Javid et al. and Amar et al., in which an increased expression of IL-17 has been described in lesional skin of DD patients[25,26]. On the one hand, the observed Th17 signature in the skin of our DD patients could be a direct effect of *ATP2A2* gene mutations in immune cells that drives Th17 skewing of naïve T cells. On the other hand, impaired epidermal homeostasis and altered microbiome could be indirectly responsible for increased expression of Th17-related genes and increased numbers of Th17 cells in the skin of DD patients. While psoriasis, as a classic Th17-dominant skin disease, is thought to be triggered at least in part by yet unknown autoantigens and not necessarily by microbial antigens, the altered microbiome of DD skin lesions with recurrent infections may promote chronic inflammation with a Th17 phenotype[26].

In summary, we demonstrated increased expression of Th17-related genes and enhanced numbers of Th17 cells in DD. We show that targeting the IL-17/IL-23 axis in a case series of three DD patients is an effective and safe therapy. One limitation of our study is the small number of patient samples, consisting of a case series involving only three treated patients. This limitation is primarily due to the scarcity of patients with DD. Moreover, the study underscores the necessity of a validated skin scoring system tailored specifically for DD, as existing scores such as the IGA fail to distinguish between inflammation activity and epidermal damage. Because DD is a chronic, relapsing disease complicated by recurrent bacterial and viral skin infections, our results might provide additional options for the long-term management of skin inflammation in patients with DD. Further randomized, controlled clinical trials are needed to evaluate the long-term benefits and side effects of this treatment modality and to clarify the formation and role of Th17 cells in DD patients.

## Methods

This study complies with all relevant ethical regulations and was approved by the ethics committee of the Medical Faculty of Johannes Kepler University Linz and the ethics committee of the University of Lausanne.

### Patients and patient samples and Molecular genetic analysis

Mutations in the ATP2A2 gene (NM_001681.4, [https://www.ncbi.nlm.nih.gov/nuccore/NM_001681]) were detected using whole exome sequencing, by enriching all exonic DNA fragments with Twist Comprehensive exome (102033) and Mitochondrial Panels (102040, both Twist Bioscience). Sequencing was performed on a NextSeq2000

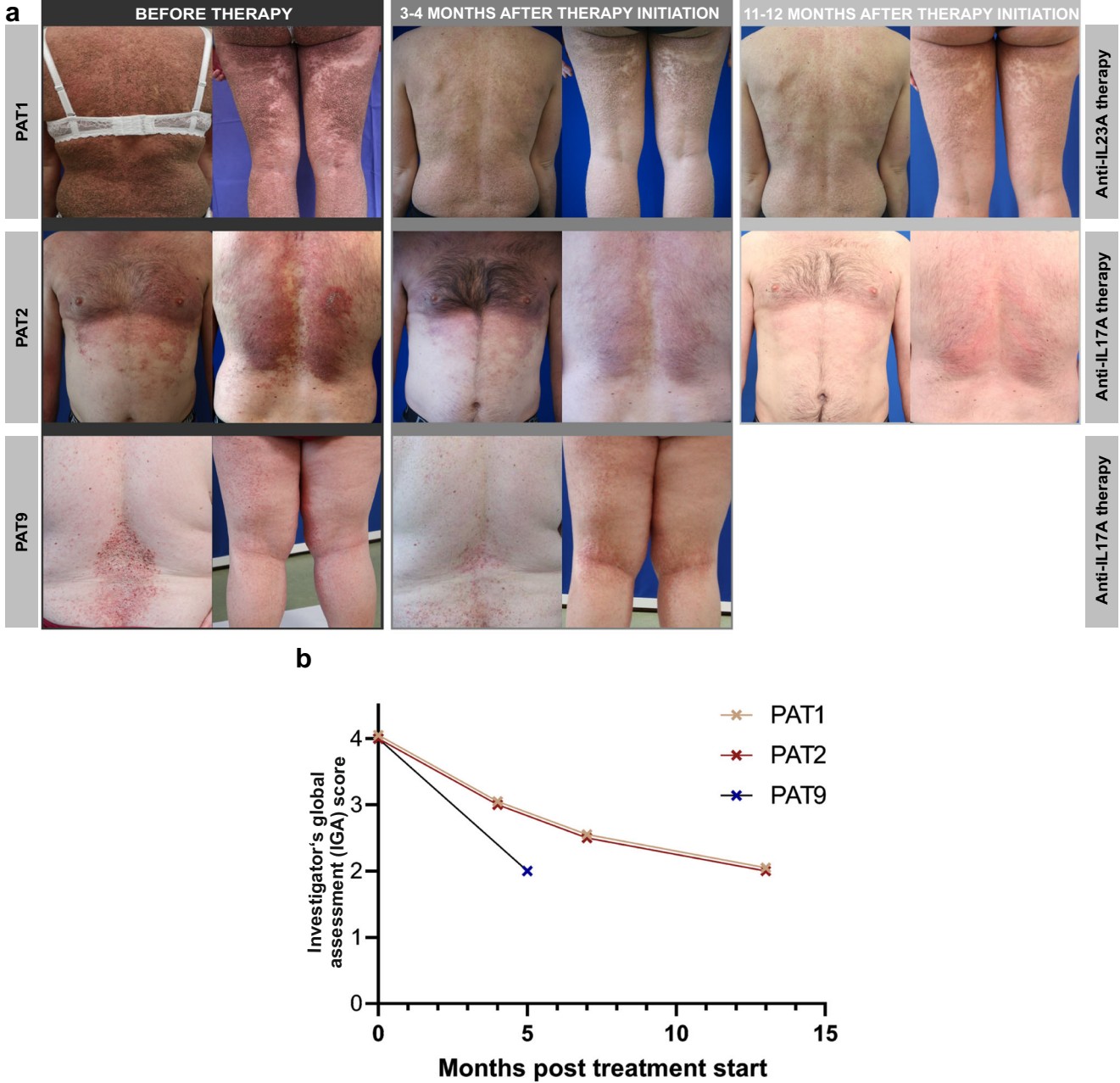

**Fig. 3 | Targeting the IL-17/IL-23 axis with monoclonal antibodies is effective in patients with Darier disease. a** Cutaneous improvement after personalized targeted antibody therapy. Clinical images of guselkumab (upper, PAT1) and secukinumab (central and lower, PAT2 and PAT9) treated patients before, 3–4 and 11–12 months after initiation of antibody therapy. **b** Clinical improvement of DD patients during therapy as determined by the Investigator's Global Assessment (IGA) score.

(Illumina) as $2 \times 150$ bp paired end reads. Mutations in the ATP2A2 gene were detected by aligning coding and flanking intronic sequences (−15/+5) of the gene with a coverage of at least 20× against the human reference sequence (GRCh37 (hg19), [https://www.ncbi.nlm.nih.gov/datasets/genome/GCF_000001405.13/]). Automatic alignment and data analysis (including copy number changes) was done using SeqNext (JSI).

**Patient samples**

Skin samples (6 mm biopsy specimens) and blood samples for collection of PBMCs were obtained from patients with Darier Disease (DD; before initiation of therapy and at specific time points during therapy), healthy controls (undergoing cosmetic surgery), and psoriasis patients (PSO) as positive controls. 6-mm biopsies were collected

on ice in RPMI medium (10.040.CV, Corning), cut into small pieces and slow frozen in FBS (10500064, Gibco) supplemented with 10% DMSO (D4540, Sigma-Aldrich) within 30 min after collection, at the most[27]. Slow frozen biopsies were used for gene and/or protein expression profiling, as described in the respective sections. PBMCs were isolated using Ficoll-Hypaque (GE17-1440-02, GE Healthcare) gradient separation from the collected blood samples. Isolated PBMCs were cryopreserved using 10% DMSO in FBS and were used for flow cytometry. Written informed consent was obtained from patients and controls. There was no monetary compensation for participation in the study. The sex of the participants was determined based on self-report. The study was approved by the ethics committee of the Medical Faculty of Johannes Kepler University Linz (EK Nr: 1327/2021, 1036/2020 and 1071/2021) and the ethics committee of the University of Lausanne

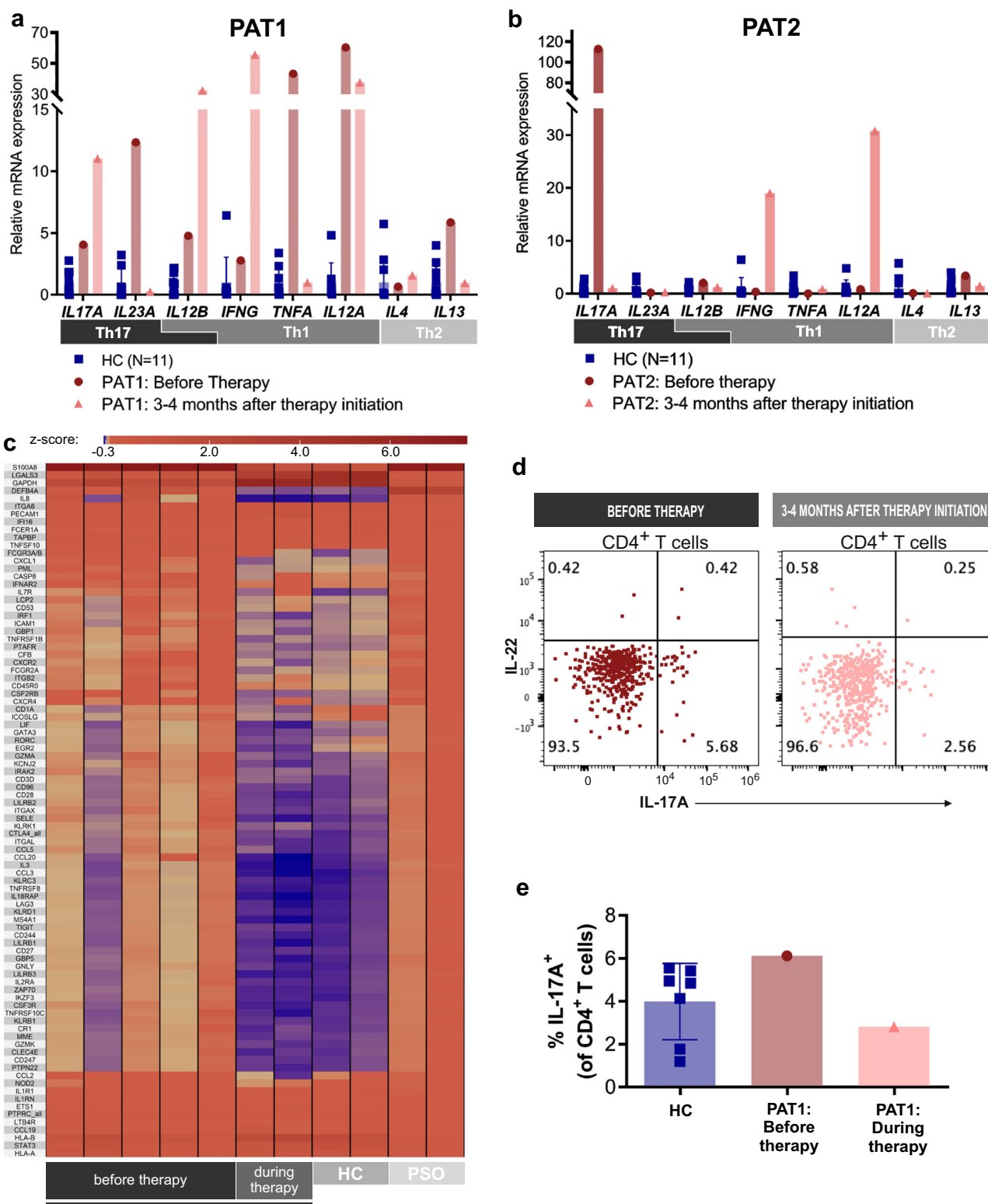

**Fig. 4 | Normalization of the inflammatory gene expression profile and Th17 cell numbers in DD patients treated with anti-IL-17/IL-23 antibodies. a, b** Relative mRNA expression (qRT-PCR) of Th1, Th2, and Th17-associated cytokines normalized to housekeeping gene ACTB and relative to HC for PAT1 and PAT2. Bars represent means of different individuals; error bars represent standard deviations. **c** Normalization of gene expression during therapy of differentially regulated genes (DD vs HC) in DD patients determined by NanoString nCounter analysis using the Immunology_V2 panel. Genes represent DE genes expressed as Z-score (DD vs HC,

fold change >1.5). **d** Assessment of Th17 cell counts with multiplex flow cytometry in skin samples from PAT1 before and during therapy with guselkumab. Dot plot of IL-22 and IL-17A expression of CD4⁺ T cells of PAT1 before therapy initiation (red) and during therapy (salmon, 3.5 months after therapy initiation). Gating strategy is shown in Suppl. Fig. 10. **e** Quantification of results in **d** compared to HC (blue, $N = 7$), PAT1 before therapy initiation (red) and PAT1 during therapy (salmon). Individual symbols represent different donors, error bars represent standard deviation.

(CER-VD 2021-00878). The study is registered with ClinicalTrials.gov (NCT05680974).

## Therapy regimen

Three DD patients (PAT1, PAT2, and PAT9) were treated off-label as part of a precision medicine approach with secukinumab (targeting IL-17A, PAT2, and PAT9) or guselkumab (targeting IL-23A, PAT1). PAT1 received guselkumab 100 mg subcutaneously at weeks 0 and 4, followed by injections every 8 weeks. PAT2 and PAT9 received secukinumab 300 mg subcutaneously once a week for 5 weeks, followed by monthly administrations.

## Clinical scores

The Investigator's Global Assessment (IGA) is a standardized severity assessment in dermatology ranging from 0 (clear) to 4 [severe (deep/dark red erythema, and marked and extensive induration/papulation; excoriation and oozing/crusting present)]. Intermediate scorings were given half points. The DLQI 10 is a 10-item questionnaire assessing health-related quality of life over the last week in patients with dermatological symptoms, with each item scored for impact from not at all (0) to very much (3) resulting in a total score ranging from 0 to 30 points. Itch was assessed by visual analog scale ranging from 0 to 10 points (0 = no pruritus, >0-< 4 points = mild pruritus, ≥4-< 7 points = moderate pruritus, ≥7-< 9 points = severe pruritus, and ≥ 9 points = very severe pruritus).

## RNA extraction

3-mm biopsies of patients' lesional skin and controls' normal skin were kept at −80 °C in RNAlater (AM7020, Invitrogen). After thawing, biopsies were mechanically disrupted using a TissueLyser II and 5-mm Stainless Steel Beads. RNA was isolated using Monarch Total Miniprep Kit (T2010S, New England Biolabs) according to the manufacturer's protocol.

## Nanostring

For multiplex gene expression profiling of the mRNA, the Nanostring nCounter System was used together with their immunology panel NS_Immunology_v2_C2328 and the nCounter SPRINT Profiler was used. Positive control was mRNA from skin lesions of seven psoriasis patients. Data was analyzed using nSolver 4.0, R and Gene Set Enrichment Analysis (GSEA)[28,29]. Unfortunately, quality control for PAT2 failed and this patient was excluded from the analysis. Data was normalized by using all housekeeping genes from the panel with an average count higher than 100 and with less than 35% coefficient of variance. The list of differentially regulated genes was generated by comparing the expression of all genes between DD and HC samples. Cut-off for fold change was 1.5 and for p-value 0.05 (Student's t test). Heatmaps were created with the R superheat package[29,30]. Data from each gene was scaled to mean 0 and standard deviation 1 (z-score).

The enrichment score (weighted Kolmogorov-Smirnov-like statistic) tells the degree to which a gene set is overrepresented at the extremes of the ranked list of compared groups (i.e., DD vs HC in Suppl. Table 2 and PSO vs HC in Suppl. Table 3), normalized enrichment score (NES) accounts for different gene set sizes and correlations between the expression data set and the gene sets. We analyzed the list of gene sets 'Biocarta Pathways' (c2.cp.biocarta.v2022.1.Hs.symbols.gmt [Curated]. We took a closer look at all gene sets with false discovery rates (FDR) lower than 0.25.

## Reverse transcriptase qPCR

cDNA Synthesis Kit (331475 L, Biozym) was used for cDNA synthesis according to the manufacturer's instructions. qRT-PCR was conducted on a LightCycler 450, using DNA Master Fast Star HybProbe (12239272001, Roche) and Taqman Gene Expression Assay (FAM) for *IL4* (Hs00174122_m1), *IL12A* (Hs01073447_m1), *IL12B* (Hs01011518_m1), *IL13*

(Hs00174379_m1), *IL17A* (Hs00174383_m1), *IL23A* (Hs00372324_m1), *TNFA* (Hs00174128_m1) and *INFG* (Hs00989291_m1, all Thermo Fisher Scientific). CT values of undetectable genes were set to 45 (lower limit of detection = number of circles recorded in qPCR experiments).

## scRNA-sequencing

Slow-frozen skin biopsies were washed with RPMI 1640 medium (61870-010, Gibco), supplemented with 10% Gold FBS (A15-151, PAA Laboratories). The skin biopsies were then minced and transferred in 1 mL of Liberase (5401119001, Roche, final concentration 0.5 mg/mL diluted with PBS without Ca and Mg) to digest the tissues. The mixture was incubated for 1 h at 37 °C and the tube was manually inverted every 15 min. Afterwards, 100 μl of 0.05% trypsin was added to the mixture and incubated at 37 °C for additional 15 min. The tissue remnants were removed by using 70 μm cell strainer. The cells were washed with MACS buffer (PBS, 0.5% BSA, and 2 mM EDTA) and then stained with SYTOX Red (S34859, ThermoFisher) and Hoechst 33342 (H3570, ThermoFisher) for cell viability assessment. The stained cells were suspended in MACS buffer and single living cells (SYTOX Red-negative and Hoechst 33342-positive) were sorted by FACS sorter (BD FACSAris-II SORP) at Flow Cytometry Facility of UNIL. Single-cell mRNA capture and sequencing were performed immediately by the Lausanne Genomic Technologies Facility (GTF) of UNIL using the Chromium Next GEM Single Cell 5′ GEM kit (10× Genomics) following the manufacturer's protocol. Raw sequencing reads were subjected to demultiplexing and aligned to the human reference genome (refdata-gex-GRCh38-2020-A) using the cellranger count tool with default parameters. The analysis included a combination of publicly available dataset (GEO portal number GSE162183)[17] comprising psoriasis (PSO) and healthy control (HC) samples, as well as an in-house dataset consisting of four DD patients. The datasets were processed together by merging the expression matrices based on common genes. To ensure data quality, cells with a read mapping rate of over 25% to mitochondrial genes, which is indicative of dying cells, were filtered out. The remaining cells were then subjected to further analysis using the Seurat package (version 4.3.0)[31]. Integration of the two datasets was performed using Harmony (version 0.1.1)[32], with the retention of the top 50 principal components. Subsequently, cell clusters were identified using the Louvain algorithm implemented in Seurat with a resolution parameter of 0.1. To assign biological annotations, known marker genes were utilized. In addition, the T cell cluster was subjected to additional clustering to distinguish T helper and cytotoxic T cell subsets.

For the differential analysis, DD and HC keratinocytes were first pseudo-bulked into three randomly assigned replicates. Then, gene expression was compared and p values were calculated using edge LRT (1) model. p-values were corrected for multiple testing using Benjamini-Hochberg method. Genes were defined as significantly overexpressed if they had a fold-change greater than 2 and a corrected p-value lower than 0.01. The gene set analysis was run using enrichR (2) on pathways from the following databases: MSigDB_Hallmark_2020, GO_Molecular_Function_2021 and SigDB_Oncogenic_Signatures, retrieving the top pathways with a p-value lower than 0.1.

## Protein expression profiling: tissue microarray and opal staining

The skin biopsies from patients were collected and FFPE histology blocks were generated with standard pathology procedures. The tissue microarray (TMA) with different type of skin diseases was created by TMA Grand Master (3DHISTECH). The immunostaining was performed with Opal Manual Detection Kit (NEL861001 KT, Akoya Biosciences) according to manufacturer's instructions. The TMA slide was incubated in two rounds of stainings; in the order of CD4 (ab133616, clone: EPR6855, Abcam, 1/250 dilution, AR9 buffer, Opal 690) and IL-17 (BS-2140R, Bioss, 1/400 dilution, AR6 buffer, Opal 520). DAPI was used as a nuclear counter stain. The stained TMA was scanned by fluorescence

scanner (PANNORAMIC 250, 3DHISTECH). The cell annotation was performed by using QuPath.

## LEGENDplex cytokine assay

For protein analysis, the skin tissues from HC and DD patients were snap frozen at −70 °C until use. Skin tissues were lysed in PBS containing Protease Inhibitor Cocktail (1:100) (P8340, Sigma-Aldrich) with the final concentration of 16–64 mg skin/ml using innuSPEED Lysis Tubes J (Cat.: 845-CS-1120250, Analytik Jena) and SpeedMill PLUS (Analytik Jena). Lysate was filtered through 0.22 μm SpinX columns (CLS8161, Sigma) and analyzed for cytokines using LEGENDplex HU Th (12-plex) assay (Cat# 741028, BioLegend). Assay samples were acquired using Cytoflex LS (Beckman Coulter) flow cytometer, using CytExpert 2.4 software. Data analysis was done using LEGENDplex Data Analysis Software Suite (QOGNIT). GraphPad prism (version 9) was used for further statistical analysis.

## Multi-color flow cytometry

For skin flow cytometry staining, biopsies were thawed, weighed and minced into smaller pieces using surgical scissors and were digested overnight with 1 ml skin digestion mix containing collagenase type 4 (0.8 mg/ml; LS004186, Worthington) and DNAse (11.77 Units/ml; D5319, Sigma-Aldrich) in RPMI-complete (RPMIc) for about 0.1 g of skin at 37 °C in 5% $CO_2$ incubator [RPMIc: RPMI 1640 (31870074, Gibco) with 5% human serum (A25761, One lambda), 1% penicillin/streptomycin (P0781, Sigma-Aldrich), 1% l-glutamine (A2916801, Gibco), 1% non-essential amino acid solution (NEAA; 11140035, Gibco), 1% sodium pyruvate (S8636, Sigma-Aldrich), and 0.1% β-mercaptoethanol (31350010, Gibco)]. After overnight digestion, the cell suspension was filtered through 70 μm mesh, then washed and resuspended in RPMIc. For detection of intracellular cytokines, skin single-cell suspensions were stimulated with phorbol 12-myristate 13-acetate (PMA) (50 ng/ml; P8139, Sigma-Aldrich) and ionomycin (1 μg/ml; I06434, Sigma-Aldrich) with brefeldin A (10 μg/ml; B6542, Sigma-Aldrich) for three hours at 37 °C in 5% $CO_2$ incubator. Cells were incubated for 15 min in Fc receptor blocking solution (422302, BioLegend). Thereafter, cells were stained in PBS containing the fixable viability dye eFluor 780, (65-0865-14, Thermo Fisher Scientific) for surface markers, using antibodies: anti-CD3 bv605, (clone SK7, 344836, BioLegend, 1/100 dilution); anti-CD4 PE-Cy5, (clone RPA-T4, 300510, BioLegend, 1/100 dilution); anti-CD45 bv510, (clone HI30, 304036, BioLegend, 1/50 dilution) and fixed and permeabilized using Foxp3 staining kit (00-5523-00, Thermo Fisher Scientific) for staining the intracellular markers using antibodies: anti-IL-17A BV786, (N49-653, 563745, BD Biosciences, 1/30 dilution); anti-IL-22 PE (clone 22URTI, 12-7229-42, Thermo Fisher Scientific, 1/80 dilution). After stimulation all the steps were done at 4 °C.

For PBMCs flow cytometry, cryopreserved PBMCs were thawed and rested overnight in RPMIc at 37 °C in 5% $CO_2$ incubator. Cells were then stimulated with PMA/ionocycin/brefeldin A followed by Fc receptor blocking and stained in PBS containing viability dye as described above for surface markers, using antibodies: anti-CD3 PE-Cy5, (clone UCHT1, 300410, BioLegend, 1/100 dilution); anti-CD3 bv421, (clone UCHT1, 300434, BioLegend, 1/100 dilution); anti-CD4 PE/Dazzle594, (clone RPA-T4, 300548, BioLegend, 1/100 dilution); anti-CD4 PE-Cy5 (clone RPA-T4, 300510, BioLegend, 1/100 dilution); anti-CD45RA AF700 (clone HI100, 304120, BioLegend, 1/50 dilution); anti-CLA bv605 (clone HECA-452, 563960, BD Biosciences, 1/50 dilution); anti-CD103 APC (BerACT8, 350216, BioLegend, 1/20 dilution). Cells were fixed and permeabilized using Cytofix/Cytoperm kit (RUO 554714, BD Biosciences) for detection of intracellular IL-17A cytokine, using antibody: anti-IL-17A PerCPcy5.5 (clone BL168, 512314, BioLegend, 1/20 dilution).

Data was acquired on Cytoflex LS (Beckman Coulter) flow cytometer, using CytExpert 2.4 software and analyzed using FlowJo software (FlowJo LLC, version10). GraphPad prism (version 9) was used for further statistical analysis.

## Reporting summary

Further information on research design is available in the Nature Portfolio Reporting Summary linked to this article.

## Data availability

The source data of this manuscript is provided as a Source Data file and are deposited in open public repositories. The Nanostring data generated in this study have been deposited in the GEO database under accession code GSE222043. The scRNAseq data of Darier's Disease patients generated in this study have been deposited in the GEO database under accession code GSE235255. The scRNA-seq of healthy and psoriasis data used is from publicly available dataset (GEO dataset GSE162183). The qPCR, FACS, and quantification of IF generated for this manuscript, as well as numerical values of clinical scores are available in the Source Data file of the manuscript. All other data are available in the article and its Supplementary files or from the corresponding author upon request. Source data are provided in this paper.

## Code availability

The code used for scRNA-seq analysis is accessible via github (https://github.com/GuenovaLab/Darrier[33].

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

## Acknowledgements
We would like to thank the patients who supported this research project. We would also like to thank all the physicians and nurses who cared for the patients. We thank Alessandra Darbellay, Sabine Köfler, Petar Noack and Ionoss Tabib for their excellent technical support. We thank Dr. Antonia Currie for proofreading the manuscript. This work was supported by grants from the Medical Faculty of the Johannes Kepler University (Linz, Austria to W.H.), the Swiss National Science Foundation (IZLIZ3_200253/1 to E.G.), the University of Lausanne (SKINTEGRITY.CH collaborative research program to E.G.), and the Forschungskredit of the University of Zurich (FK-15-040 to W.H.).

## Author contributions
M.E., S.K., and W.H. planned the study, revised the data, and wrote the manuscript. T.B. processed samples from the clinics, performed qPCR, NanoString, and Immunofluorecence experiments, Y.T.C. and Y.C.T. performed the IL-17 staining and the scRNA-seq experiment, A.S. and S.R.V. performed FACS and bead-based cytokine detection experiments, P.P. was responsible for the scRNA-seq analysis, L.C.S. established and performed bead-based cytokine detection experiments, C.I. performed TMA assembly and analysis, M.E., I.D., J.T., R.L., and G.W. provided samples of patients and controls and clinical data, A.L. processed clinical samples and performed qPCR. S.K. analyzed the data and assembled the figures. M.E., I.K.G., S.A., E.G., and W.H. revised the data and the manuscript.

## Competing interests
The authors declare no competing interests.
