## [Peer Review File · Nature Communications]

Increased expression of the Th17-associated cytokines IL-17 and IL-23 in inflamed skin of Darier disease patients as novel therapeutic targetsREVIEWER COMMENTS

Reviewer #1 (Remarks to the Author):

Ettinger et al. present a manuscript on targeting Th17 cytokines in patients with Darier Disease, a rare autosomal dominant genodermatoses. The authors studied six patients and performed immunoprofiling of skin samples by NanoString technology. They suggest that the expression profile resembles psoriasis. They found an increase in IL-17A expression compared to healthy skin. Then they treated one patient with an IL-23 inhibitor and a second patient with an IL-17A inhibitor and report from a decrease in disease severity. Although it is interesting to learn more about this rare disease there are several major concerns with this report. Darier Disease is a genodermatoses and mutations in ATP2A2 are well described. Also the pathogenesis due to altered calcium ATPase pump, SERCA2 and ER stress and the development of acantholysis and cell apoptosis are studied in detail. All these processes are located in the epidermis. The authors studied whole skin samples.

The epidermal alterations in Darier Disease facilitate secondary infections with bacteria, viruses and fungi. The seborrheic body areas are typically affected. It is not surprising to find some IL-17A and related factor expression in superinfected skin. But this is different from a pathogenic role of a Th17 immune response as responsible for psoriasis. The immune cells found in skin pathology from Darier's patients are secondary to the superinfection and are not attracted by ATP2A2 mutations. At least the authors don't provide any evidence for this. The concept of the study does not consider at all these facts.

Figure 1.

A Statistical power is missing. The authors included only two healthy control samples and two psoriasis samples. There is no way for statistical testing.

B Why do the authors focus on immune pathways only? Is this a selection of pathways identified?

C It is not correct to use a statistical test for three groups with n=2 (psoriasis). Also psoriasis skin does not express any IL-4. The log scale is misleading for a comparison of Darier Disease with psoriasis. The authors should use a linear scale.

D Immunofluorescence shows no specific staining for IL-17-producing T cells and controls are missing.

Figure 2.

A The clinical course in patients with Darier Disease shows variations and improvements can occur over time, while summer period or UV exposure can worsen the clinics. Also the use of ointments can dramatically change the disease severity. It is unclear which cofactors may have influenced the disease severity of the two patients described here.

B Why do the authors use a different set of genes for the heat map than in figure 1?

C If the light red and red bars represent single patients (patient 1 or patient 2), why are several symbols depicted and how is it possible to use statistical testing if only one patient is studied? There is no Turkey's post hoc test.

Reviewer #2 (Remarks to the Author):

This is a study of 6 patients with DD confirmed by histology and genetics. Immunoprofiling was performed on these patients. This revealed increased IL 17 signaling specifically IL17A. This was further tested with IHC staining of skin biopsies in these patients.

Patient 1 was treated with guselkumab and patient 2 was treated with secukinumab and noted improvement in symptoms.

This is a nice paper exploring the immune pathogenesis of DD and proof of conception for targeted immune treatment.

Reviewer #3 (Remarks to the Author):

Prospective study of New treatment for Darier Dz

Authors

A case series of 6 DD patients

5 with inc IL17, one IL23

Treatment with sekukinumab & guselkumab improved signs & biopsy expression of cytokines

1. Why hasn't a validated disease extent score or IGA score been used?
2. Why not a QOL or itch tool?
3. Where is the dose and frequency of injection given? What was the rationale for the

dosing?

4. Ethics?

Over the past several months, we performed all of the additional experiments requested by the reviewers, and we are now in the position to address all of the concerns raised by the reviewers.

Reviewer 1:

Ettinger et al. present a manuscript on targeting Th17 cytokines in patients with Darier Disease, a rare autosomal dominant genodermatoses. The authors studied six patients and performed immunoprofiling of skin samples by NanoString technology. They suggest that the expression profile resembles psoriasis. They found an increase in IL-17A expression compared to healthy skin. Then they treated one patient with an IL-23 inhibitor and a second patient with an IL-17A inhibitor and report from a decrease in disease severity.

Although it is interesting to learn more about this rare disease there are several major concerns with this report. Darier Disease is a genodermatoses and mutations in ATP2A2 are well described. Also the pathogenesis due to altered calcium ATPase pump, SERCA2 and ER stress and the development of acantholysis and cell apoptosis are studied in detail. All these processes are located in the epidermis. The authors studied whole skin samples.

The epidermal alterations in Darier Disease facilitate secondary infections with bacteria, viruses and fungi. The seborrheic body areas are typically affected. It is not surprising to find some IL-17A and related factor expression in superinfected skin. But this is different from a pathogenic role of a Th17 immune response as responsible for psoriasis. The immune cells found in skin pathology from Darier's patients are secondary to the superinfection and are not attracted by ATP2A2 mutations. At least the authors don't provide any evidence for this. The concept of the study does not consider at all these facts.

We thank the reviewer for this important point. We agree that effects of the disease-causing mutations in the ATP2A2 gene has been well studied in the epidermis, where they lead to aberrant Ca²⁺ signaling, loss of intercellular connections (acantholysis) and cell apoptosis in keratinocytes. We now provide additional data on gene expression profiles in keratinocytes of DD patients compared with healthy control skin by scRNA-seq (Fig. 1A, Suppl. Fig. 3, Suppl. Table 1, page 3, lines 32-34 & page 4, lines 1-10). On the other hand, ATP2A2 gene mutations occur in all cells of the body, and epidemiological studies have shown an increased risk of neuropsychiatric disorders, type 1 diabetes, and heart failure in DD patients. As a result, DD has recently been termed a multi-organ disease. Given the chronic cutaneous inflammation in DD patients, we thoroughly investigated the composition and phenotype of the immune cells in DD skin lesions. Based on additional scRNA-seq experiments and multi-color immunofluorescence stainings of tissue micro array sections, we now demonstrate the presence of Th17 cells as the main source

of IL-17 in DD skin (Fig. 1A-C, Suppl. Fig. 5, Suppl. Fig. 6, page 4, lines 23-34 & page 5, lines 1-8). Finally, we prove that blocking the IL-23/IL-17 axis with monoclonal antibodies is of clinical relevance, as this therapeutic approach markedly improves skin inflammation and clinical scores (IGA, itch score, DLQI) in patients. These data strongly suggest that Th17 cells play an important role in the pathogenesis of DD that has not been previously described. Whether the formation of Th17 cells in DD skin lesions is an indirect effect attributable to the altered microbiome with recurrent skin infections in patients or whether ATP2A2 gene mutations in T cells might directly drive naïve T cells towards a Th17 phenotype remains unclear. In preliminary *in vitro* experiments, we observed an enhanced skewing of naïve T cells isolated from the blood of DD patients towards a Th17 phenotype compared to naïve T cells obtained from healthy donors (Fig. 1 below). These data suggest a direct effect of ATP2A2 gene mutations in T cells, likely mediated via altered Ca^{2+} signaling in the formation of Th17 cells. Further research is needed to decipher the role of ATP2A2 gene mutations in T cells and its effect on the formation of the different Th phenotypes. However, these experiments are beyond the scope of this study. According to the reviewer's suggestion, we have rewritten parts of the discussion section (page 6, lines 4-13 & 30-34).

Fig. 1: Enhanced formation of Th17 cells in DD patients. Naïve CD4⁺ T cells were isolated from blood of DD patients (N=4) and healthy donors (N=4) (Miltenyi bead assay). T cells were incubated for 7 days under Th17 skewing conditions (cytokine mix: IL-6, IL-21, IL-23, TGF β and IL-1 β , protocol: Nat Immunol. 2018;19(10):1126-1136). Cells were then restimulated with PMA/Ionomycin and analyzed by flow cytometry for the indicated cytokines. *P>0.05

Figure 1A Statistical power is missing. The authors included only two healthy control samples and two psoriasis samples. There is no way for statistical testing?

Thank you for this important comment. We agree with the reviewer that we cannot draw any statistically significant conclusions from the Nanostring analysis. However, in the next experiments, we directly confirmed the Nanostring data with regard to the IL-17 expression with qRT-PCR using skin samples from 6 DD patients, 7 psoriasis patients and 11 healthy controls.

Figure 1B Why do the authors focus on immune pathways only? Is this a selection of pathways identified?

Given the chronic skin inflammation in DD patients, the focus of our study was to investigate the composition and phenotype of the immune cells in DD skin lesions. Therefore, we chose a Nanostring chip (Immunology_V2 panel) that primarily covers cytokine and immunology-related genes. Consequently, pathway analysis focused on immune pathways without preselection of specific Biocarta pathways.

Figure 1C It is not correct to use a statistical test for three groups with n=2 (psoriasis). Also psoriasis skin does not express any IL-4. The log scale is misleading for a comparison of Darier Disease with psoriasis. The authors should use a linear scale.

We thank the reviewer for these important points. We increased the number of psoriasis patients and healthy controls according to the reviewer's suggestion. In total, we now provide cytokine expression data measured by qRT-PCR of 6 DD patients, 7 psoriasis patients and 11 healthy controls (Fig. 1D). We fully agree with the reviewer that psoriasis does not express IL-4. Because of the larger sample size, IL-4 expression now matches the levels found in healthy control skin (Fig. 1D). Furthermore, we have changed the log scale to a linear scale as suggested by the reviewer (Fig. 1D).

Figure D Immunofluorescence shows no specific staining for IL-17-producing T cells and controls are missing.

We thank the reviewer for this point. We have reanalyzed the immunofluorescence staining and now provide additional experiments on IL-17 producing cells. To identify the origin of IL-17 in lesional skin of DD patients, we analyzed IL17 gene expression at the single cell level by scRNA-seq and additionally performed multicolor immunofluorescence staining of tissue microarray sections. We detected IL-17A/F expression predominantly in the Th cell subset in skin samples of DD and psoriasis (Fig. 2A-C) compared with healthy skin, indicating that Th17 cells are the main

source of IL-17 in inflamed skin of patients with DD. These new results have been added to the manuscript text (page 4, lines 23-34 & page 5, lines 1-8). In addition, we now provide new immunofluorescence images of Th17 cells and have included the corresponding isotype controls (Suppl. Fig. 6).

Figure 2A The clinical course in patients with Darier Disease shows variations and improvements can occur over time, while summer period or UV exposure can worsen the clinics. Also the use of ointments can dramatically change the disease severity. It is unclear which cofactors may have influenced the disease severity of the two patients described here.

We fully agree with the reviewer. As mentioned in the introduction, DD is characterized by a chronic relapsing course with exacerbations triggered by various factors (e.g. sun exposure, heat, friction, or infection). In our study, we successfully treated three patients with severe, therapy-resistant DD (Table 1) with IL-23 or IL-17 blocking antibodies. Two of the three patients have already received therapy over one year now without relapse or flares. In detail, we observed a continuous improvement of skin manifestations and clinical scores (IGA, itch VAS, DLQI) in our patient cohort. In our opinion, the observed improvement with a stable course of disease without relapse over such a long period is the result of specific antibody therapy. Extensive treatment of the skin with topical corticosteroid or antiseptics was not allowed. In addition, clinical improvement was accompanied by normalization of the inflammatory gene expression profile and a reduction of Th17 cells in skin samples taken during therapy (Fig. 3 D-E).

Figure 2B Why do the authors use a different set of genes for the heat map than in figure 1?

The set of genes was not changed. However, because the genes are ordered based on hierarchical clustering, the order of the genes in Fig. 3D (old Fig. 2) was eventually different compared the Fig. 1B. We thank the reviewer for this comment. We have changed the order of the genes in Fig. 3 according to Fig. 1.

Figure 2C If the light red and red bars represent single patients (patient 1 or patient 2), why are several symbols depicted and how is it possible to use statistical testing if only one patient is studied? There is no Turkey's post hoc test.

Multiple data points represent experimental replicates of the data, i.e., we performed qPCR twice (each time in triplicate), using RNA from the same biopsy. We agree with the reviewer that our

approach is statistically incorrect. We changed the graphs accordingly and additionally omitted statistical testing (new Fig. 3C). We would like to thank the reviewer for pointing out the typo in Tukey's post hoc test.

Reviewer 2:

This is a study of 6 patients with DD confirmed by histology and genetics. Immunoprofiling was performed on these patients. This revealed increased IL 17 signaling specifically IL17A. This was further tested with IHC staining of skin biopsies in these patients. Patient 1 was treated with guselkumab and patient 2 was treated with secukinumab and noted improvement in symptoms. This is a nice paper exploring the immune pathogenesis of DD and proof of conception for targeted immune treatment.

We thank the reviewer for this comment.

Reviewer 3:

Why hasn't a validated disease extent score or IGA score been used?

We thank the reviewer for this important comment. We documented disease progression during therapy with the IGA score as suggested. The IGA assessment showed a 50% reduction in the IGA score from 4 points (severe disease) at baseline to 2 points (mild disease) during treatment. We have added the new data in Fig. 3B and included the results in the manuscript text (page 5, lines 16-18).

Why not a QOL or itch tool?

In parallel to the IGA scoring, we assessed itch with a VAS score and measured quality of life with the DLQI. We observed a reduction in itch and an improvement of the patients' quality of life. The results are shown in Suppl. Fig. 8.

Where is the dose and frequency of injection given? What was the rationale for the dosing?

Three patients (PAT1, PAT2 and PAT9) were treated with secukinumab (targeting IL-17) or guselkumab (targeting IL-23), respectively. PAT1 received guselkumab 100 mg subcutaneously at weeks 0 and 4, followed by injections every 8 weeks, and PAT2 and PAT9 received secukinumab 300 mg subcutaneously once a week for 5 weeks, followed by monthly

administrations. The frequency and dose of injections was in line with the FDA approved therapy scheme for psoriasis. The injections were administered in the outpatient clinic. We have included a detailed description of the dosing scheme in the Method section of our manuscript (page 13, lines 26-31).

Ethics?

The study was approved by the ethics committee of the Medical Faculty of Johannes Kepler University Linz (EK Nr: 1327/2021, 1036/2020 and 1071/2021; collection of patient samples, analyses of patient samples, off-label therapy) and the ethics committee of the University of Lausanne (CER-VD 2021-00878; scRNA-seq, tissue microarray). Furthermore, the off-label therapy of DD patients with biologics based on the individual cytokine analyses is registered with ClinicalTrials.gov (NCT05680974). The information has been added to the method section of the manuscript (page 13, lines 9-24).

REVIEWERS' COMMENTS

Reviewer #1 (Remarks to the Author):

This is a revised version of the paper by Ettinger et al. on targeting Th17 related cytokines in patients with Darier disease. The authors added additional data to the original submission and made changes to the text. They now agree that they have used incorrect statistical testing before. The authors tryd to address the referees concerns. The additional data improved the manuscript.

However, they still overstate their findings. Especially, the discussion has to be written more modest. This is a case study with two to three patients, who received treatment with monoclonal antibodies. As illustrated in Fig 3A the clinical improvements are not highly impressive. If IL-17 has a major role in this genetic disorder – as suggested by the authors – the clinical improvement should be visible more clear and more rapid. There is no histological follow up.

The title of the paper should be changed. They used anti-IL-17A and anti-IL-23 antibodies in single patients. There is no use of any other Th17-related cytokine targeting. They should not give the impression that this is a clinical study or trial.

In the abstract the sentence 'We prove.....` should be rephrased. The paper does not contain enough data to prove any efficacy or safety of a treatment. Three patients have been treated, two with anti-IL-17A and one with anti-IL-23.

Discussion section. The authors totally ignore the paper by Javid et al. published in Clin Exp Dermatol in 2023 demonstrating IL-17A expression in the skin of 7 patients with Darier disease. They should cite and discuss this publication. Also, the discussion section should be rephrased, since the expression of IL-17A and related genes in Darier disease has been demonstrated earlier (PMID: 36632755).

Ref. 23 - what journal/issue?

Fig. 1 According to the data demonstrated in figure 1 there is no or minimal expression of

IL23A. Why should this cytokine be important in Darier disease if there is no expression found? In addition, the authors did not detect IL23A expression in psoriasis samples. Any explanation?

Fig. 3 The patient treated with anti-IL-23A antibody shows an increase in IL-17A expression although the authors observed clinical improvement. This IL-17 expression result contradicts the author's main suggestion that IL-17 is relevant in the pathogenesis of Darier disease.

Reviewer #3 (Remarks to the Author):

The revised version of the manuscript has improved the evidence to suggest that IL17 and IL23 inhibitors may improve Darier disease with a scientific rationale despite this being a genodermatosis.

The photographs are compelling.

The IGA score provided just mentioned levels such as 'severe' without a descriptor for each level 0-4 defining each one. Hence, the graphs do not reflect what is seen in the photographs.

In order to gain approval for a biologic in any skin disease, a validated skin score is required. For other blistering type skin diseases it has been useful to separate activity from damage such as pigmentation- eg PDAI, BPDAI.

A weakness of this paper and aim for the future would be to develop and validate a DD clinical severity score and IGA score with a group of experts. This is vital for trials to proceed. I would be happy to assist with this important step.

Dedee F Murrell

Reviewer 1:

This is a revised version of the paper by Ettinger et al. on targeting Th17 related cytokines in patients with Darier disease. The authors added additional data to the original submission and made changes to the text. They now agree that they have used incorrect statistical testing before. The authors tried to address the referees concerns. The additional data improved the manuscript.

However, they still overstate their findings. Especially, the discussion has to be written more modest. This is a case study with two to three patients, who received treatment with monoclonal antibodies. As illustrated in Fig 3A the clinical improvements are not highly impressive. If IL-17 has a major role in this genetic disorder – as suggested by the authors – the clinical improvement should be visible more clear and more rapid. There is no histological follow up.

We appreciate the reviewer for highlighting this crucial point. In response, we have made the following revisions:

- Modified the title.
- Rewritten sections of the discussion (page 6, line 6; page 6, lines 20-23; page 7, lines 1-3;).
- Included the limitations of our study in the discussion section (page 7, lines 14-20).
- Additionally, we have explicitly mentioned in the manuscript that we conducted testing of monoclonal antibody therapy on three patients as part of a case series.

The title of the paper should be changed. They used anti-IL-17A and anti-IL-23 antibodies in single patients. There is no use of any other Th17-related cytokine targeting. They should not give the impression that this is a clinical study or trial.

We have changed the title of our manuscript to “Increased expression of the Th17-associated cytokines IL-17 and IL-23 in inflamed skin of Darier disease patients as novel therapeutic targets” (page 1, lines 1-3).

In the abstract the sentence ‘We prove.....’ should be rephrased. The paper does not contain enough data to prove any efficacy or safety of a treatment. Three patients have been treated, two with anti-IL-17A and one with anti-IL-23.

We agree with the reviewer and have rephrased the sentence ‘We prove.....’ in the abstract (page 2, lines 11-12).

Discussion section. The authors totally ignore the paper by Javid et al. published in Clin Exp Dermatol in 2023 demonstrating IL-17A expression in the skin of 7 patients with Darier disease. They should

cite and discuss this publication. Also, the discussion section should be rephrased, since the expression of IL-17A and related genes in Darier disease has been demonstrated earlier (PMID: 36632755).

We thank the reviewer for this important point. In line with our observation, two recent publications have reported an elevated IL-17 signature in genetic acantholytic and blistering disorders including DD (Clin Exp Dermatol 27;48(5):518 and Microbiome 26;11(1):162). We have incorporated these references into our list and have integrated their findings into the discussion section (page 7, lines 1-3).

Ref. 23 - what journal/issue?

We appreciate the reviewer for bringing this error to our attention. We have included the journal name and issue number of this specific reference in the reference section (page 16).

Fig. 1 According to the data demonstrated in figure 1 there is no or minimal expression of IL23A. Why should this cytokine be important in Darier disease if there is no expression found? In addition, the authors did not detect IL23A expression in psoriasis samples. Any explanation?

We fully agree with the reviewer. In our qRT-PCR and scRNA-seq analyses of psoriasis skin samples compared to healthy control skin, we did not detect increased levels of IL23A. Therefore, we assessed IL-23 protein expression directly using multi-color immunofluorescence stainings of tissue microarray (TMA) sections (Fig. 1). Quantification of TMA stainings revealed increased IL-23A expression in PSO and DD samples compared to AD (atopic dermatitis), which is known to be a classical Th2-(IL-4/IL-13) dominated skin disease (Fig. 1). We have included this new dataset in Fig. 2.

Fig. 1: IL-23A/CD4 staining of skin tissue micro array. Left panel: Representative multi-color immunofluorescence (IF) images of tissue micro array sections stained with OPAL technique for IL-23A, CD4 and DAPI in skin samples of DD patients (N=7), atopic dermatitis patients (AD, N=3) and PSO (N=3). CD4 positive T cells are marked in red, IL-23A positive cells are marked in green, DAPI-stained nuclei are marked in blue. Right panel: Quantification of IL-23A-positive cells as determined by mean fluorescence intensity of multi-color immunofluorescence stained TMA sections (skin samples: DD (N=7, red), PSO (N=3, beige), AD (N=3, blue)).

Fig. 3 The patient treated with anti-IL-23A antibody shows an increase in IL-17A expression although the authors observed clinical improvement. This IL-17 expression result contradicts the author's main suggestion that IL-17 is relevant in the pathogenesis of Darier disease.

We fully agree with the reviewer. In the qRT-PCR analysis of skin biopsies taken three months after therapy initiation with an anti-IL-23A antibody, PAT1 showed an increase in IL17A expression. Because mRNA expression in bulk tissue RNA may differ from actual protein expression, we directly assessed Th17 cell numbers (IL-17⁺CD4⁺ T cells) using multiplex flow cytometry in skin biopsies before and during therapy. As shown in Fig. 4E (formerly Fig. 3E), we observed increased numbers of Th17 cells in lesional DD skin compared to healthy control skin before therapy. Of importance, clinical improvement was associated with a decrease in Th17 cell numbers to normal control levels (healthy skin) during therapy with an anti-IL-23A antibody (Fig. 4F).

Reviewer 3:

The revised version of the manuscript has improved the evidence to suggest that IL17 and IL23 inhibitors may improve Darier disease with a scientific rationale despite this being a genodermatosis.

The photographs are compelling.

The IGA score provided just mentioned levels such as 'severe' without a descriptor for each level 0-4 defining each one. Hence, the graphs do not reflect what is seen in the photographs.

In order to gain approval for a biologic in any skin disease, a validated skin score is required. For other blistering type skin diseases it has been useful to separate activity from damage such as pigmentation- eg PDAI, BPDAI.

A weakness of this paper and aim for the future would be to develop and validate a DD clinical severity score and IGA score with a group of experts. This is vital for trials to proceed. I would be happy to assist with this important step.

We thank the reviewer for this important comment. We totally agree with the reviewer that a validated skin score, specifically tailored for DD, is essential for future clinical trials. In line with the reviewer's recommendation, we plan to develop and validate such a score with a panel of experts, utilizing a larger patient cohort in the near future. We have also incorporated the reviewer's comment into the limitations section of our study, found on page 7, lines 14-20.